# Perceived stress may mediate the relationship between antenatal depressive symptoms and preterm birth: A pilot observational cohort study

**Sharifa Lalani**[1], **Aliyah Dosani** [2,3]*, **Ntonghanwah Forcheh** [4], **Shahirose Sadrudin Premji** [4], **Sana Siddiqui**[5‡], **Kiran Shaikh**[1‡], **Ayesha Mian**[5], **Ilona S. Yim**[6], **the Maternal-infant Global Health Team (MiGHT) Collaborators in Research**¶

1 School of Nursing and Midwifery, Aga Khan University, Karachi, Pakistan, 2 School of Nursing and Midwifery, Mount Royal University, Calgary, Alberta, Canada, 3 Department of Community Health Sciences, University of Calgary, Calgary, Alberta, Canada, 4 School of Nursing York, University, Toronto, Ontario, Canada, 5 Department of Psychiatry, Aga Khan University, Karachi, Pakistan, 6 Department of Psychological Science, University of California, Irvine, Irvine, California, United States of America

☯ These authors contributed equally to this work.
‡ These authors also contributed equally to this work.
¶ Membership of the Maternal-infant Global Health Team is provided in the Acknowledgments.
* adosani@mtroyal.ca

**Data Availability Statement:** The data underlying this study contain sensitive participant information and cannot be shared publicly. Please contact

## Abstract

### Background

Screening for changes in pregnancy-related anxiety and depressive symptoms during pregnancy may further our understanding of the relationship between these two variables and preterm birth.

### Objectives

To determine whether changes in pregnancy-related anxiety and depressive symptoms during pregnancy influence the risk of preterm birth among Pakistani women; explore whether perceived stress moderates or mediates this relationship, and examine the relationship between the various components of pregnancy-related anxiety and preterm birth.

### Methods

A prospective cohort study design was used to recruit a diverse sample of 300 low-risk pregnant women from four centers of Aga Khan Hospital for Women and Children in Karachi, Pakistan. Changes in pregnancy-related anxiety and depressive symptoms during pregnancy were tested. Multiple logistic regression analysis was used to determine a predictive model for preterm birth. We then determined if the influence of perceived stress could moderate or mediate the effect of depressive symptoms on preterm birth.

Saima Ejaz, member of the Aga Khan University Institutional Review Board with further data inquiries (saima.ejaz@aku.edu).

**Funding:** Canadian Institutes of Health Research (Planning Grant Number 264531; Project Grant Number 376731-PJT. SSP received the funding) and Aga Khan University, University Research Council Multi-Disciplinary Project (Grant Number 144005SONAM. SK received the funding). The funders had no role in study design, data collection and analysis, decision to publish, or preparation of the manuscript.

**Competing interests:** The authors have declared that no competing interests exist.

## Results

Changes in pregnancy-related anxiety (OR = 1.1, CI 0.97–1.17, p = 0.167) and depressive symptoms (OR = 0.9, CI 0.85–1.03, p = 0.179) were insignificant as predictors of preterm birth after adjusting for the effects of maternal education and family type. When perceived stress was added into the model, we found that changes in depressive symptoms became marginally significant after adjusting for covariates (OR = 0.9, CI 0.82–1.01, p = 0.082). After adjusting for the mediation effect of change in perceived stress, the effect of change in depressive symptoms on preterm birth were marginally significant after adjusting for covariates. Among six different dimensions of pregnancy-related anxiety, mother's concerns about fetal health showed a trend towards being predictive of preterm birth (OR = 1.3, CI 0.97–1.72, p = 0.078).

## Conclusions

There may be a relationship between perceived stress and antenatal depressive symptoms and preterm birth. This is the first study of its kind to be conducted in Pakistan. Further research is required to validate these results.

## Introduction

Preterm birth is commonly defined as birth before 37 weeks' gestation [1]. The global incidence of preterm birth is estimated at 15 million per year with an average of 11.8% of births being preterm in low-income countries [2]. Pakistan has a particularly high burden of preterm birth (18.9%) [3], which exceeds those of other countries in the region including India (15%), Bangladesh (11%), and Indonesia (15.5%) [4–6]. As such, preterm birth is a global public health concern since it contributes directly to neonatal mortality and childhood morbidity [7]. Thus, addressing preterm birth is critical to addressing neonatal and child mortality and morbidity, particularly in resource-poor settings.

Pregnancy-related anxiety, antenatal depressive symptoms, and perceived stress have been identified as risk factors for adverse maternal-infant birth outcomes [8–13]. Over the last two decades, many studies, including a meta-analysis [14] have shown that higher pregnancy-related anxiety is associated with preterm birth. However, the nuances in this relationship remain to be elucidated. While Dole and colleagues [15] found that women experiencing medium and high counts of pregnancy-related anxiety items showed an increased risk of preterm birth (RR = 1.5, 95% CI 1.1–2.1; RR = 2.1, 95% CI 1.5–3.0), Orr and colleagues [16] found this to be true of high counts of pregnancy-related anxiety only (OR 1.50–2.73, 95% CI 1.01–7.27). Similarly, Kramer and colleagues [17] were able to demonstrate that pregnancy-related anxiety had a dose-response relationship with spontaneous preterm birth (OR = 1.8, 95% CI 1.3–2.4). Although both Rauchfuss and Maier [18] and Tomfohr-Madsen and colleagues [19] found that pregnancy-related anxiety was positively associated with preterm birth (OR 1.44, 95% CI 1.02–2.05; OR 8.54), Tomfohr-Madsen and colleagues [19] found that shorter sleep duration had a moderating role in the relationship between pregnancy anxiety and birth outcomes. Further research is required to understand the relationship between the various components of the pregnancy-related anxiety scale and preterm birth in different settings. Conversely, antenatal depressive symptoms were associated with preterm birth in many [8, 20–25], but not all [26–28] studies. For instance, Liu and colleagues [29] found that

mothers who experienced both new and recurrent depression during pregnancy were more likely to give birth early (OR = 1.34, 95% CI 1.22–1.46; OR = 1.42, 95% CI 1.32–1.53). In their population-based study, Li and colleagues [21] found that women who experienced increasing severity of depression were more likely to give birth early, suggesting a potential dose-response relationship. However, Fransson and colleagues [20] concluded that even moderate levels of depressive symptoms significantly elevated the risk for preterm birth (OR = 3.14, 95% CI 1.37–7.19). Furthermore, Grote and colleagues [30] found that while women with depression during pregnancy are at increased risk for experiencing a preterm birth, the magnitude of the effect size varies depending on how it is measured, country of residence, and socioeconomic status. Conversely, not all of the available studies have found similar relationships. For example, Gavin and colleagues [26] found no association between depressive symptoms and preterm birth respectively (OR = 1.1, 95% CI 0.6–1.9). The literature is also inconsistent with respect to the relationship between perceived stress and preterm birth. Dole and colleagues [15] and Seravalli and colleagues [31] have demonstrated a trending towards an association between perceived stress and preterm birth (RR = 1.3, 95% CI 0.9–1.8; OR = 1.49, 95% CI 1.00–2.23 [14, 28, 29] while Krabbendam and colleagues [32] and Sealy-Jefferson and colleagues [33] have identified no effect (OR = 1.10, 95% CI 0.77–1.59; PR = 1.14, 95% CI 0.97–1.34). Therefore, a more complex relationship may exist between antenatal depressive symptoms, pregnancy-related anxiety, perceived stress, and preterm birth.

The inconsistencies in the literature may be due to a failure to consider changes in pregnancy-related anxiety and antenatal depressive symptoms over the course of pregnancy, and its association to preterm birth. As such, changes in measures of psychosocial distress may be more informative than assessments at a single time point in understanding the dynamic relationship between perinatal distress and adverse pregnancy outcomes. A few authors have found that there may be a blunting of psychological and biological responses to perceived stress late in the second trimester, which may protect the mother and the fetus from adverse health outcomes [13, 34, 35]. Pregnant women who did not perceive a decrease in perceptions of stress levels over the course of pregnancy or dampening of biological responses into the late second trimester, could be at increased risk of preterm birth [34, 36].

A few studies have examined the relationship between psychosocial distress and preterm birth by considering the dynamic nature of pregnancy. These studies have repeatedly assessed measures of pregnancy-related anxiety [13, 37, 38] and antenatal depressive symptoms [27, 39–43] in the second and third trimester. Doktorchik and colleagues [39] studied the relationship between changes in anxiety and depression between 17 to 24 weeks' of gestation and 32 to 36 weeks' of gestation with preterm birth. They found that women who experienced an increase in anxiety scores had 170% higher odds of preterm birth compared to those who had a decrease in anxiety scores. Other studies have also repeatedly assessed measures of pregnancy-related anxiety [13, 27, 37, 38] and antenatal depressive symptoms [40, 42] in the second and third trimester. Doktorchik and colleagues [39] found a similar pattern with anxiety, indicating that women who experience an increase with anxiety scores over the course of pregnancy were at greater risk for preterm birth in Canada. The same pattern did not hold true for antenatal depressive symptoms. While the overall risk for prematurity increased if women experienced more than one psychological disorder—pregnancy-related anxiety, antenatal depressive symptoms, and perceived stress [41], Doktorchik and colleagues [39] found that a co-occurring increase in anxiety and depression scores did not increase the risk of preterm birth, and perceived stress did not modify any of these relationships.

It is also the case that empirical research critically examining psychosocial processes predominantly originates from high-income countries [44–46] and fails to recognize the complex multisystem interactions in the pathways to adverse health outcomes for both mother and

fetus over the course of pregnancy [47]. The viewpoint of a life course suggests that early life experiences and repetitive stressors over an individual's life will lead over time to "wear and tear" on the brain and body, undermining psychosocial and biological responses to stress [48, 49]. These biopsychological responses in early life manifest later as psychological health issues during adulthood and pregnancy thereby contributing to preterm birth [50]. On the contrary, our recent work does not support the relationship between adverse life experiences and preterm birth in the Pakistani context [51].

In low- and middle-income countries, pregnant women face disparities in health determinants (e.g. social, cultural, economic, and political contexts) that can change perceptions of psychosocial distress and biological responses, thereby differentially affecting psychological and biological response patterns during pregnancy [47]. At the societal level, women in Pakistan are more vulnerable to adverse events and life course alterations [47]. Individual situations such as maternal employment and support systems are other notable factors influencing maternal psychological health. For instance, when fathers are not involved during the pregnancy the risk of preterm birth was found to be 21% higher (OR = 1.21, 95% CI: 1.01–1.45) [52, 53]. Women working strenuous jobs are two times more likely to deliver preterm babies as compared to those performing management tasks and sedentary work [53, 54]. Correspondingly, more than 50% of women with unplanned pregnancies experience higher stress levels opposed to women with planned pregnancies [55]. Women experiencing neighborhood disadvantage in terms of physical disorder in the neighborhood and racism, particularly black women, were also two times more likely to deliver preterm (OR = 2.64; OR = 2.16, 95% CI 1.39–3.35) [53, 56, 57]. The interplay between the multitude of risk factors on preterm birth has not been fully explored, particularly in low- and middle-income countries where likelihood of adverse health outcomes is far greater [58]. Therefore, exploration of the relationship between antenatal depressive symptoms, pregnancy-related anxiety, perceived stress, and preterm birth is needed to understand the possible contribution on birth outcomes.

The objectives of this study were to use data from a resource-poor country to: (a) determine whether changes in pregnancy-related anxiety and antenatal depressive symptoms during pregnancy influence the risk of preterm birth; (b) explore whether perceived stress influences the relationship between changes in pregnancy-related anxiety and antenatal depressive symptoms and preterm birth; (c) to determine if perceived stress moderates or mediates the relationship between antenatal depressive symptoms, pregnancy-related anxiety, and preterm birth; and (d) examine the relationship between the various components of the pregnancy-related anxiety scale [59] and preterm birth.

## Materials and method

### Study design and setting

This is a secondary analysis of data that were collected between October 2015 and July 2016. A diverse sample of 300 low risk pregnant women were recruited from four centers of Aga Khan Hospital for Women and Children in Pakistan, including Hyderabad, Garden, Kharadar, and Karimabad, using a prospective cohort design (Fig 1). Aga Khan Hospital for Women and Children is a university-affiliated teaching hospital with 8,000 deliveries per year, with laboratory facilities at each location.

### Study procedures and participants

Ethical approval was received from the Ethics Research Committee (ERC) of the Aga Khan University Hospital (ID: 3564-SON-ERC-15) and the Conjoint Health Research Ethics Board of University of Calgary (ID: REB15-2372). The study was conducted in accordance with the

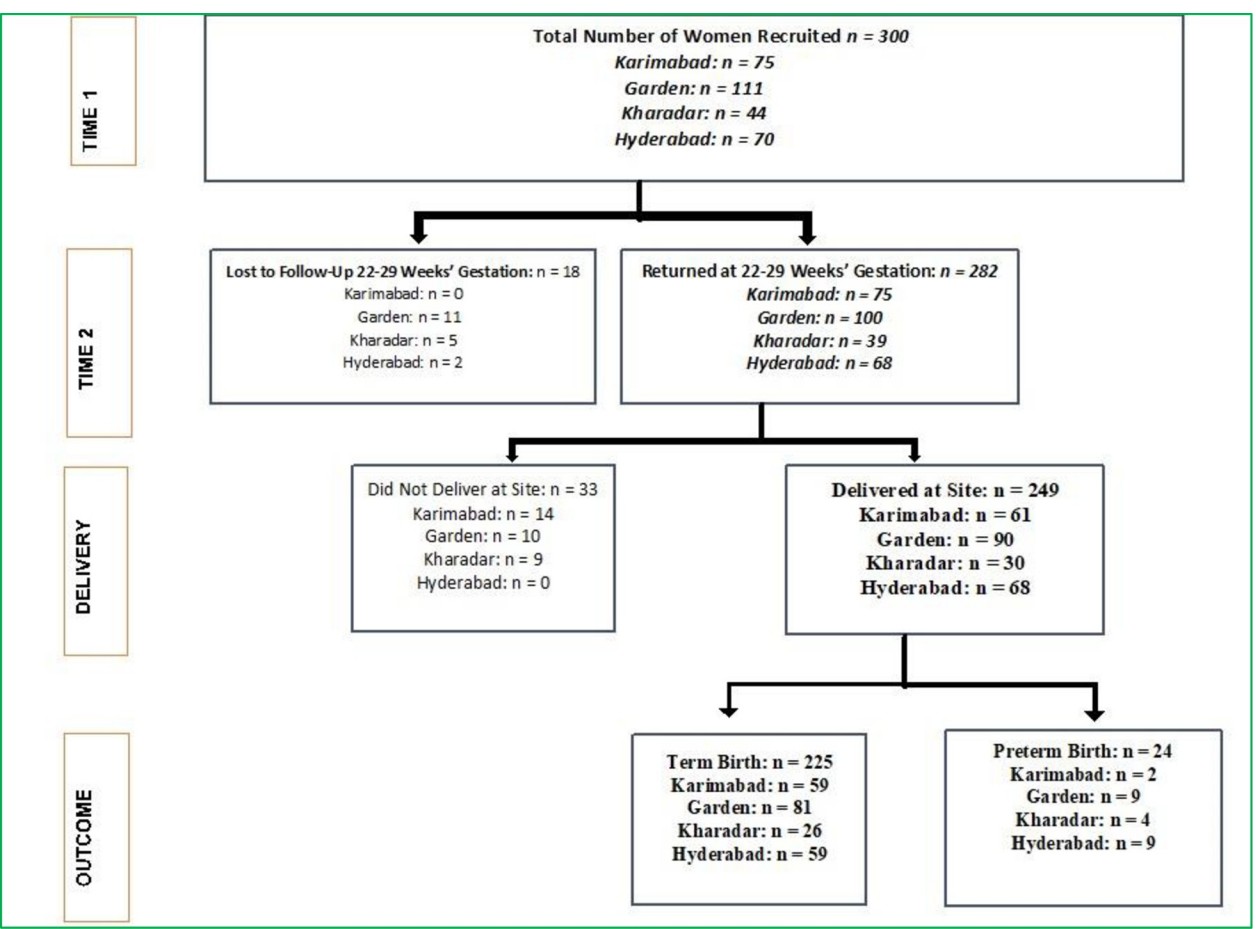

**Fig 1. Data flow.**

Declaration of Helsinki. Women who were 18 years or older, were able to speak Urdu or English, and whose pregnancy was within 12–19 weeks' gestational age as determined by the last menstrual cycle were included in our study. Women with self-reported pregnancy co-morbidities such as hypertension and diabetes mellitus were excluded. A detailed description of our exclusion criteria is reported elsewhere [51]. Written informed consent to participate in the study was obtained from all women.

## Participant recruitment, data collection, and measurement

Fig 1 shows the recruitment and loss to follow-up numbers. There was significant variation in sample sizes by location ranging from 90 (36%) women from Garden to 30 (12%) from Kharadar (Fig 1). Women were scheduled for an initial visit and a follow-up visit 10 weeks after recruitment. Text message reminders were sent to encourage women to attend follow-up appointments. Sociodemographic variables were assessed at the first visit to obtain information about possible socio-demographic covariates, including age, ethnicity, highest education attained, age at marriage, family type (nuclear or extended), years of marriage, number of children ever born, work status, and household income. A total of 36 items were explored in the demographic survey.

Instruments for measuring the three main predictor variables, namely pregnancy-related anxiety, antenatal depressive symptoms, and perceived stress were administered during recruitment in the early second trimester (time 1 = 12–19 weeks' gestational age) and 10 weeks later in the late second trimester (time 2 = 22–29 weeks' gestational age). Of the 300 women, 282 (94%) returned for follow up about 10 weeks after enrolment and provided updated data on pregnancy-related anxiety, antenatal depressive symptoms, and perceived stress. Of these, 83% women (n = 249) returned to deliver at their respective clinics, thus constituting the analytic sample for this study (249/300 = 83%). Change scores for antenatal depressive symptoms and pregnancy-related anxiety were computed by subtracting the total score obtained at the first visit from the total score obtained at the second visit. Birth outcome data were collected for all women who returned for delivery at their recruitment center.

## Predictor variables

The Pregnancy-Related Anxiety Scale [60], is a 10-item, 4-point scale. Each item was scored from 0 representing no anxiety to 3 representing high anxiety. Cronbach's α for the scale is 0.78. The six subscales for this scale represent: anxiety about childbirth, confidence/control, anxiety about fetal health, anxiety about loss of fetus, mother's personal wellbeing and parenting. The first five items are scored on a scale of 1 to 4 while items 6–10 are rated as never, sometimes, most of the time, or all of the time. While there is no established cut-off for the scale, three or more positive responses in column four has been used to suggest the presence of anxiety.

The 10-item Edinburgh Postnatal Depression Scale [61, 62], a 4-point Likert scale was administered at recruitment and later at follow-up to measure depressive symptoms. This tool has been validated and commonly used in pregnant and postpartum samples. Items 1 to 4 are scored from 0–3, while items 5–10 are scored from 3–0. The numbers are then tallied. Using a cut-off score of 13 across 15 countries, the Edinburgh Postnatal Depression Scale has a Cronbach's α = 0.73–0.87 [61–65].

## Mediating/Moderating variable

The 10-item Perceived Stress Scale [66] a 5-point Likert scale instrument was used to compute a measure of perceived stress. Each item was scored from 0 (no perceived stress) to 4 (high perceived stress). Perceived stress scores are obtained by reversing responses (e.g., 0 = 4, 1 = 3, 2 = 2, 3 = 1 & 4 = 0) to the four positively stated items (items 4, 5, 7, & 8) and then summing across all scale items. With a cut-off score of 20, the Perceived Stress Scale has a Cronbach's α = 0.78–0.91 [66–68].

## Outcome variable

Preterm birth, the occurrence of birth prior to 37 weeks' gestation, was determined using last menstrual cycle, based on participant self-report at recruitment.

## Data analysis and statistical methods

SPSS V25 was used for analysis and statistical modelling. Descriptive data was explored with continuous variables reported as means and standard deviation and categorical variables as frequency and percentages. Exploratory investigations into the patterns of pregnancy-related anxiety, antenatal depressive symptoms, and perceived stress scores over two time-periods using paired t-tests and Pearson correlations were completed with the assumption of normality investigated. Hierarchical multiple logistic regression was used to obtain a predictive model

for preterm birth given changes in pregnancy-related anxiety and antenatal depressive symptoms with and without adjustment for significant covariates. Assumptions of hierarchical multiple regression modeling were investigated. For the hierarchical model, conditional forward likelihood criterion with the default probability of inclusion = 0.05 and exclusion = 0.10 was used to determine significant covariates, and changes in depressive symptoms and pregnancy-related anxiety scores were then added to the most parsimonious model and their significance evaluated. We refitted each of the models by adjusting for perceived stress and evaluated the corresponding changes in predictor variables parameters and their significance. We then conducted a process analysis to determine if the influence of perceived stress could be moderating or mediating the effect of depressive symptoms on preterm birth. We use Hayes process model 1 with covariates for the moderation analysis and model 4 with covariates for mediation analysis [69, 70].

Finally, overall pregnancy-related anxiety was separated into its composite parts to determine which components of pregnancy-related anxiety would emerge as significant predictors of preterm birth. We investigated if any of the six individual subscales of pregnancy-related anxiety, along with depressive symptoms, influenced the risk of preterm birth and, whether perceived stress influenced this relationship. Thus, changes in overall pregnancy-related anxiety were replaced with changes in each of the six subscales. The model was then refitted, and forward conditional likelihood criterion used to enter the most significant factors. We then adjusted the model for perceived stress and then adjusted for the covariates.

## Results

### Descriptive data

Descriptive data are presented in Table 1. Of the 249 women, 24 gave birth preterm (9.6%). In terms of age, 2.3% were under 20 years old, while 30.1%, 41.0%, and 26.1% of the women were aged 20–24, 25–29 and 30+ years, respectively. The largest ethnic group was the Muhajirs (30.5%), followed by Sindhi (19.7%) and Memon (14.1%). No other ethnic group represented more than 5% of the participants and were therefore combined into a single group. Almost half (49.8%) of the women were aged 20–24 years at time of first marriage, while 18.9% where aged 16–19 years, and the remaining were aged 25–36 years old. Most women (74.3%) lived with extended family members, 39.8% were pregnant for the first time, 36.2% already had another child (20.1% had a boy, 16.1% had a girl), and 24.1% had two or more other children. Household income was over 40K Pakistani Rupees for 31.3% of our sample, 20-40K Pakistani Rupees for 32.1% of the women, and 12.0% were from households earning 10K Pakistani Rupees or less. More than half of the women (54.6%) had post-secondary qualifications, including 18.5% with postgraduate degrees while 23.3% had primary or no formal education. Hence, most women were from affluent households and well-educated. The mean baseline score, change in means (D), standard error (SE) and two-sided paired t-tests, and p-value (p) for pregnancy-related anxiety, antenatal depressive symptoms, and perceived stress were 11.8 (D = 0.418, SE = 0.328, p = 0.204), 5.9 (D = -0.932, SE 0.324, p = 0.004), and 15.1 (D = -0.386, SE = 0.433, p = 0.375) respectively.

### Patterns of pregnancy-related anxiety, antenatal depressive symptoms, and perceived stress during pregnancy

An EPDS score $\geq$ 12 was classified as moderate/severe depressive symptoms. We found 11.2% of our study participants experienced moderate/severe depressive symptoms at baseline and 17.3% at follow-up. In addition, 5.3% of study participants experienced moderate/severe

**Table 1. Preterm birth rate by sociodemographic characteristics.**

| Sociodemographic Characteristics | | Percent | (% Preterm birth) | n | p-value |
|---|---|---|---|---|---|
| Total sample | All women | 100.0 | (9.6) | 249 | |
| Study site | Garden | 36.1 | (10.0) | 90 | 0.228 |
| | Hyderabad | 27.3 | (13.2) | 68 | |
| | Karimabad | 24.5 | (3.3) | 61 | |
| | Kharadar | 12.0 | (13.3) | 30 | |
| Mother's highest education attained | Primary education or None | 23.3 | (19.0) | 58 | 0.027 |
| | Secondary or high school | 22.1 | (7.3) | 55 | |
| | College/University completed | 36.1 | (8.9) | 90 | |
| | Postgraduate degree | 18.5 | (2.2) | 46 | |
| Mother's ethnic group | Muhajir | 30.5 | (5.3) | 76 | 0.023 |
| | Sindhi | 19.7 | (2.0) | 49 | |
| | Memon | 14.1 | (14.3) | 35 | |
| | Other | 35.7 | (15.7) | 89 | |
| Age at marriage | Under 20yrs | 18.9 | (14.9) | 47 | 0.298 |
| | 20–24Yrs | 49.8 | (9.7) | 124 | |
| | Over 24Yrs | 31.3 | (6.4) | 78 | |
| Family type | Nuclei | 25.7 | (17.2) | 64 | 0.018 |
| | Extended | 74.3 | (7.0) | 185 | |
| Years of marriage | Under 2yrs | 29.7 | (9.5) | 74 | 0.337 |
| | 2–4 | 32.5 | (6.2) | 81 | |
| | 5+ | 37.8 | (12.8) | 94 | |
| Number of children | First Child | 39.8 | (9.1) | 99 | 0.14 |
| | One Boy | 20.1 | (4.0) | 50 | |
| | One Girl | 16.1 | (7.5) | 40 | |
| | Two or more children | 24.1 | (16.7) | 60 | |
| Work status | Homemaker | 79.5 | (10.1) | 198 | 0.626 |
| | Other | 20.5 | (7.8) | 51 | |
| Household income | <Rs10,000 | 12.0 | (13.3) | 30 | |
| | 10,001–20,000 | 24.5 | (8.2) | 61 | 0.881 |
| | 20,001–40,000 | 32.1 | (10.0) | 80 | |
| | >Rs40,000 | 31.3 | (9.0) | 78 | |

depressive symptoms at both baseline and follow-up. Participants experiencing any mental health concerns were offered additional mental health care, free of charge. Specifically, all four data collection sites offer psychiatric services. Women reporting stress, anxiety, and depression, including those who were experiencing suicidal thoughts were offered a free referral to psychiatrist. The psychiatrist provides counselling services and prescribes medication if needed. In addition, we created another pathway to accessing care where women who required additional mental health support could have been referred to one of the co-investigators, a clinical psychologist, regardless of their decision to continue or withdraw from the study. However, none of our study participants took up the additional mental health support that was offered. Exploratory analysis revealed that there was a small but statistically significant increase in antenatal depressive symptoms (mean increase = 0.93, p = 0.004) between time 1 and 2, but no overall change in pregnancy-related anxiety (p = 0.204) or perceived stress (p = 0.375). The Pearson correlation coefficients between the scores at time 1 and corresponding scores at time 2 were positive, moderate in magnitude and statistically significant

**Table 2. Predictive model for preterm birth.**

| Factors | Categories | Unadjusted Model with Predictors Only | | Model with Only Covariates* | | Model Adjusted for Significant Covariates | |
|---|---|---|---|---|---|---|---|
| | | p-value | OR (95% CI) | p-value | OR (95% CI) | p-value | OR (95% CI) |
| Mother's highest education attained | | | | 0.057 | | 0.083 | 0 |
| | Primary education or None | | | 0.026 | 10.8 (1.33–88.33) | 0.031 | 10.1 (1.23–83.46) |
| | Secondary or high school | | | 0.245 | 3.8 (0.40–35.41) | 0.214 | 4.2 (0.44–39.65) |
| | College/University completed | | | 0.168 | 4.4 (0.53–36.97) | 0.189 | 4.2 (0.50–35.07) |
| | Postgraduate degree (reference) | | | | 1 | 0.000 | 1 |
| Family type | Nuclear family | | | 0.023 | 2.8 (1.15–6.73) | 0.043 | 2.5 (1.03–6.25) |
| | Extended (reference) | | | | 1 | 0.000 | 1 |
| Pregnancy-related anxiety | (Unit increase) | 0.062 | 1.1 (1.00–1.19) | | | 0.167 | 1.1 (0.97–1.17) |
| Depressive symptoms | (Unit increase) | 0.076 | 0.9 (0.84–1.01) | | | 0.179 | 0.9 (0.85–1.03) |
| | Constant | | 0.100 | | 0.015 | | |

*Parsimonious model, excluding non-significant covariates using variable selection.

(pregnancy-related anxiety: r = 0.446 p < 0.001; antenatal depressive symptoms: r = 0.56, p < 0.001 and perceived stress: r = 0.34, p < 0.001).

**Research question 1: Do changes in pregnancy-related anxiety and antenatal depressive symptoms during pregnancy in the Pakistani context influence the risk of having a preterm birth?** In Table 2, the unadjusted model, the model with all of the covariates included, and the adjusted model are presented. The unadjusted model for preterm birth revealed a trend towards significance, suggesting that there is a relationship between pregnancy-related anxiety (p = 0.062) and depressive symptoms (p = 0.076) and preterm birth. In the model where all covariates were included, mother's highest education attained (p = 0.057) and family type (p = 0.023) were retained in the covariates model. Hence, these two covariates were used to adjust the model. While mother's education overall was trending towards significance, the women with the lowest level of education were significantly different from those with a postgraduate degree. Change in both pregnancy-related anxiety (p = 0.167) and depressive symptoms (p = 0.179) did not emerge as significant predictors of preterm birth after adjusting for the effects of the two significant covariates. The adjusted R-squared for the final model was 11.8% indicating a low predictive power.

**Research question 2: Does perceived stress influence the relationship between changes in pregnancy-related anxiety and antenatal depressive symptoms and preterm birth?** The results of the three models shown in Table 3 correspond to the three models in Table 2, each adjusted for perceived stress. Findings in Table 3 show that the change in perceived stress from time 1 to time 2 was not a significant predictor of preterm birth (p = 0.168). However, changes in perceived stress had a significant influence on the effect of changes in depressive symptoms on preterm birth. When the model with predictors only was adjusted for changes in perceived stress, changes in depressive symptoms emerged as a significant predictor of preterm birth (p = 0.026) whereas changes in pregnancy-related anxiety remained a non-significant trend (p = 0.089). An increase in the level of depressive symptoms by 1, led to a decrease in the odds of having preterm birth by a factor of 0.89 (95% CI = 0.80–0.99) for fixed level of change in pregnancy-related anxiety. When the model with stress and predictors was adjusted for covariates (Table 3) the effect of depressive symptoms became only marginally significant (p = 0.082), while the effect of change in pregnancy-related anxiety remained insignificant (p = 0.214). These results indicate that change in perceived stress has some protective influence

**Table 3. Predictive model for preterm birth given changes in pregnancy-related anxiety and antenatal depressive symptoms during pregnancy adjusted for perceived stress.**

| Factors | Categories | Model Adjusted for Perceived Stress Only | | Model with Covariates and Perceived Stress Only | | Model with Predictors Adjusted for Covariates and Perceived Stress | |
|---|---|---|---|---|---|---|---|
| | | p-value | OR (95% CI) | p-value | OR (95% CI) | p-value | OR (95% CI) |
| Mother's highest education attained | | | | 0.051 | | 0.079 | |
| | Primary education or None | | | 0.026 | 10.8(1.32–88.36) | 0.037 | 9.5(1.14–78.24) |
| | Secondary or high school | | | 0.269 | 3.6(0.38–33.36) | 0.271 | 3.6(0.37–34.04) |
| | College/University completed | | | 0.186 | 4.2(0.5–34.98) | 0.235 | 3.6(0.43–30.82) |
| | Postgraduate degree (Ref) | | | | | | 1.0 |
| Family type | Nuclear family | | | 0.023 | 2.8(1.15–6.73) | 0.050 | 2.5(1.00–6.16) |
| | Extended (reference) | | | | 1.0 | | 1.0 |
| Perceived stress (unit increase) | | 0.168 | 1.1(0.98–1.14) | 0.406 | 1.0(0.96–1.11) | 0.198 | 1.1(0.97–1.15) |
| Pregnancy-related anxiety (unit increase) | | 0.089 | 1.1(0.99–1.18) | | | 0.214 | 1.1(0.97–1.17) |
| Depressive symptoms (unit increase) | | 0.026 | 0.9(0.80–0.99) | | | 0.082 | 0.9(0.82–1.01) |
| | Constant | | 0.102 | | 0.02 | | 0.02 |

on the relationship between the change in depressive symptoms and preterm birth, but not on the effect of change in pregnancy-related anxiety. Furthermore, the predictive ability of change in pregnancy-related anxiety on preterm birth is marginal after controlling for mother's education and family type. The adjusted R-squared for the final model was 12.4% indicating a low predictive power.

**Research question 3: Does perceived stress moderate or mediate the relationship between antenatal depressive symptoms and preterm birth?** While the change in depressive symptoms was only marginally predictive of preterm birth after adjusting for the covariates, our analyses indicated that change in perceived stress did have an influence. In the moderation model in Table 4, $b_1$ is the conditional effect of depressive symptoms when change in perceived stress is 0 for fixed values of the covariate (education and family type). This effect is only marginally significant (p = 0.080). Similarly, $b_2$ is the effect of change in perceived stress when there is no change in depressive symptoms between time 1 and 2 and is only marginally significant (p = 0.090). The interaction effect between change in perceived stress and change in depressive symptoms measures the moderation effect of perceived stress on depressive symptoms. The estimate, ($b_3$ = 0.0, p = 0.51) is not significant, indicating that change in perceived stress is not a moderator of the possible effect of change in depressive symptoms on preterm birth.

The mediation analysis shown in Table 5 indicates that there is a significant positive relationship between change in depressive symptoms and change in perceived stress (a = 0.68,

**Table 4. Regression analysis examining the moderation of the effect of change in depressive symptoms on preterm birth by change in perceived stress.**

| | Role | Symbol | Coefficient | Standard Error | p-Value |
|---|---|---|---|---|---|
| Change in depressive symptoms (X) | Predictor | $b_1$ | -0.09 | 0.05 | 0.080 |
| Change in perceived stress (W) | Mediator | $b_2$ | 0.07 | 0.04 | 0.090 |
| Interaction (XW) | Interaction | $b_3$ | 0.00 | 0.01 | 0.510 |
| Nuclear family (C1) | Covariate | $g_1$ | -0.93 | 0.46 | 0.050 |
| No education | Covariate | $g_2$ | 1.15 | 0.47 | 0.010 |
| Constant | Constant | $i_y$ | -1.91 | 0.40 | p<0.001 |
| | | | Model $\chi^2$ (5) = 15.88, p = 0.01 | | |

**Table 5. Regression analysis examining the mediation of the effect of change in depressive symptoms on preterm birth by change in perceived stress.**

| Antecedent | Role | | Mediator (Perceived Stress) | | | | Outcome (Preterm Birth) | | |
|---|---|---|---|---|---|---|---|---|---|
| | | | Coefficient | Standard error | p- value | | Coefficient | Standard error | p-value |
| Depressive symptoms | Predictor | a | 0.68 | 0.07 | < 0.001 | c' | -0.1 | 0.05 | 0.070 |
| Perceived stress | Mediator | - | | | | b | 0.07 | 0.04 | 0.100 |
| Nuclear family | Covariate | $f_1$ | 0.19 | 0.86 | 0.830 | $g_1$ | -0.99 | 0.46 | 0.030 |
| Education | Covariate | $f_2$ | -0.76 | 0.89 | 0.400 | $g_2$ | 1.21 | 0.46 | 0.010 |
| Constant | Constant | $i_m$ | -2.10 | 0.23 | <0.001 | $i_y$ | -1.95 | 0.39 | <0.001 |
| | | | | | $R^2 = 0.26$ | | | X by M interaction: $\chi^2(1) = 0.5$, p = 0.48 | |
| | | | | | $F(3,245) = 29.19$, p<0.001 | | | Model $\chi^2(4) = 15.88$, p<0.001 | |

p < 0.001), suggesting that perceived stress affects the relationship between depressive symptoms and other variables. The mediation analysis reveals that both the direct effect, c' = -0.10, p = 0.07 and indirect effect, ab = 0.040 (CI: 0.00–0.10) of change in depressive symptoms on preterm birth, adjusting for the mediation effect of change in perceived stress are only marginally significant after adjusting for covariates.

**Research question 4: What is the relationship between the various components of the pregnancy-related anxiety scale and preterm birth?** Although the overall change in pregnancy-related anxiety was not associated with preterm birth, we found that one of its dimensions, 'Concerns/worries about Fetal Health', emerged as a significant predictor (p = 0.031) of preterm birth (Table 4). A unit of increase in concerns about fetal health led to an increase of 1.4 (95% CI: 1.03–1.78) times in the odds of preterm birth, and these odds remained essentially unchanged (OR = 1.3) after controlling for changes in perceived stress (Table 6). In addition, the effects of depressive symptoms on preterm birth now emerged as significant (p = 0.026). A unit increase in depressive symptoms led to a reduction in the odds of preterm birth by 0.9 (95% CI 0.81–0.99). When the model was adjusted for perceived stress and covariates (Table 6), both depressive symptoms (p = 0.085) and concerns about fetal health (p = 0.078) became only marginally insignificant. This should be interpreted with caution, since the model became unstable as more predictors were added due to the small number of women with preterm births (n = 24). Notwithstanding this limitation, the overall pattern of results provides a

**Table 6. Predictive model for preterm birth given changes in dimensions of pregnancy-related anxiety and depressive symptoms during pregnancy adjusted for perceived stress and covariates.**

| Factors | Categories | Model with Predictors Only | | Model Adjusted for Perceived Stress | | Model Adjusted for Perceived Stress and Covariates | |
|---|---|---|---|---|---|---|---|
| | | P-value | OR (95% CI) | p-value | OR (95% CI) | p-value | OR (95% CI) |
| Mother's highest education attained | | | | | | 0.055 | |
| | Primary education or None | | | | | 0.030 | 10.4(1.25–86.86) |
| | Secondary or high school | | | | | 0.293 | 3.4(0.35–31.85) |
| | College/University completed | | | | | 0.216 | 3.9(0.45–32.71) |
| | Postgraduate degree (ref) | | | | | | 1 |
| Family type | Nuclear | | | | | 0.067 | 2.4(0.94–5.9) |
| | Extended (Ref) | | | | | | 1 |
| | Perceived stress | | | 0.198 | 1.1(0.97–1.13) | 0.210 | 1.1(0.97–1.14) |
| | Pregnancy-related Anxiety: fetal health concerns | 0.031 | 1.4(1.03–1.78) | 0.050 | 1.3(1.00–1.75) | 0.078 | 1.3(0.97–1.72) |
| | Depressive symptoms | 0.070 | 0.9(0.84–1.01) | 0.026 | 0.9(0.81–0.99) | 0.085 | 0.9(0.82–1.01) |
| | Constant | | 0.10 | | 0.10 | | 0.02 |

strong indication that perceived stress has an impact on the relationship between changes in antenatal depressive symptoms and preterm birth.

To summarize the study results, the findings reveal that two significant covariates mother's education and family type does not influence change in pregnancy-related anxiety and depressive symptoms and are not significant predictors of preterm birth. Also, change in perceived stress has protective influence on the relationship between change in depressive symptoms and preterm birth but not on effect of change in pregnancy-related anxiety. Moreover, the predictive ability of change in pregnancy-related anxiety and preterm birth is marginal after controlling two significant covariates that is mother's education and family type.

The mediation analysis shows that there is a significant positive relationship between change in depressive symptoms and change in perceived stress. Mother's concerns/worries about fetal health', emerged as a significant predictor of preterm birth. The above results provide a strong indication that perceived stress has an impact on the relationship between changes in antenatal depressive symptoms and preterm birth.

## Discussion

The strongest and most consistent predictors of preterm birth in our sample were mother's level of education and family type. Women with higher educational attainment and those living with extended families showed a significantly lower rate of preterm birth. Our results are consistent with others who propose that educated women are more likely to look for, understand, and follow medical advice about optimal behavior during pregnancy [71, 72] and social support from partners, family and friends acts as a buffer to prevent negative consequences from stress, including preterm birth [52, 53, 73].

In contrast, the change in pregnancy-related anxiety was not a predictor after adjusting for covariates and further adjusting for perceived stress. This result is different from that obtained by Weis and colleagues [74] who found that a 1/10 unit rise in the anxiety slope related to accepting pregnancy, labor fears, and helplessness increased the risk of preterm birth by 37%, 60%, and 54%, respectively. Dissimilarities in results may be attributed to the different tools used. Weis and colleagues [74] measured pregnancy-specific maternal anxiety using the 53-item, Lederman Prenatal Self-Evaluation Questionnaire-Short Form (PSEQ-SF), while we used the pregnancy-related anxiety scale. In our research, while the pregnancy-related anxiety subscale 'concerns about fetal health' initially emerged as a significant predictor, its effect became insignificant with adjustment for covariates and perceived stress. The predictive ability of change in depressive symptoms was significant after adjusting for perceived stress but became marginally significant after further adjusting for covariates. The influence of the change in depressive symptoms on preterm birth, after adjusting for the mediation effect of change in perceived stress is marginally significant after adjusting for covariates. None of the other more recent studies that collected data on perinatal distress at more than one time point measured the impact of the change in level of distress and its impact on preterm birth [19, 43]. To our knowledge, this is the first study of its kind to be conducted in Pakistan.

The results of this study are similar to Doktorchik and colleagues [39] who found that perceived stress did not modify any of the relationships between changes in pregnancy-related anxiety and antenatal depression and preterm birth. This study found that perceived stress did have a mediation effect on the changes in antenatal depressive symptoms that was marginally significant after adjusting for covariates. However, it is important to note that we focused on anxiety specifically related to the pregnancy whereas Doktorchik and colleagues [39] assessed anxiety broadly, using the Spielberger State Anxiety Scale [75]. In addition, the covariates that Doktorchik and colleagues [39] used to adjust their model (i.e. social support, maternal age at

delivery, ethnicity, income and having a history of preterm birth) were different than the covariates used to adjust the model in this study (education and family type).

Our results differ from those of Glynn and colleagues [13] who found that pregnant women who exhibit a decrease in perceptions of psychosocial distress and dampening of biological stress responses in the late second trimester were less likely to deliver preterm. Glynn and colleagues [13] found that there may be a blunting of psychological and biological responses to perceived stress late in the second trimester, which may protect the mother and the fetus from adverse health outcomes. While Glynn and colleagues [13] found that both changes in perceived stress and pregnancy related anxiety contributed to preterm birth, they did not consider the impacts of both changes in perceived stress and changes in pregnancy-related anxiety in the same model. The results of this study were different and could have been due to considering changes in antenatal depression and changes in perceived stress in the same model in addition to the moderation effects of the changes in perceived stress.

Our analysis revealed that when covariates (maternal education and family type) were added to the model, and the model further adjusted for perceived stress, the effect of depressive symptoms became only marginally significant. This indicates that perceived stress has some effect on preterm birth and that the unique predictive value of depressive symptoms on preterm birth overlaps with the predictive value of perceived stress on preterm birth. Various antenatal stressors, including general anxiety and stressful life events can work together to increase the risk of antenatal depression [76]. In Pakistan, women have been found to experience inequities in access to education, and employment opportunities and also, have severe life stressors including absolute poverty and limited social resources that impact the ability for women to seek the care they require [47]. A focus on these life stressors may take away from the perceived stress of pregnancy, thereby increasing chance of preterm birth.

We found that women with a decrease in perceptions of psychosocial distress in the late second trimester have higher odds of preterm birth. Decreases in the perception of pregnancy-related anxiety, or a focus on other needs outside of the pregnancy may have led to decreased needed attention and thereby risk of preterm birth. It is difficult to comment on the definitive reasons why we are seeing this correlation; a certain element of stress is functional in nature and allows the mother to take care of the needs of pregnancy [13]. In addition, there may also be a biological response when the body adapts to stressors to maintain allostasis or stability [48]. However, while these mediators can be protective or adaptive, overuse (allostatic load) or wear and tear can result in dysregulation or organ system failure, producing harmful effects, such as preterm birth [77, 78].

Perceived stress can be significant for Pakistani women as exposure to social, cultural, and environmental phenomena including poverty, neighborhood threats, intimate partner violence, and childhood hardships further compound inequities experienced and has been associated with preterm birth [43, 79]. It should be noted, however, that although Pakistan is a low-to-middle income country up to half of the women in our sample had university education and were attending private clinics. Furthermore, prevalence and risk profiles for antenatal depressive symptoms vary across cultures [80] and other factors may be responsible for the differences than what is currently found in some of the cited literature, indicating that further research is required.

Women in low- and middle-income countries may experience, and report perceived stress, pregnancy-related anxiety, and depressive symptoms differently from those covered in previous studies, due to cultural factors that limit the ability of women to have a voice in decisions that impact their health [78]. For example, in the South Asian environment, gender inequity and social acceptance of domestic violence may contextualize the experience of pregnancy-related anxiety [81]. Hobel and colleagues [82] contend that there are two factors that are

relevant to the risk of preterm birth: not only is the timing of the stressor significant, but the woman's perception of the perceived stressor is equally important.

When we analyzed our data with the six components of pregnancy-related anxiety separately [59], we found that fetal health was the only significant predictor of preterm birth, with one unit of increased concerns about fetal health leading to 1.4 times the odds of having preterm birth. This association remained when adjusted for perceived stress. However, this association became marginally significant after adjusting for covariates. Differences in sociocultural situations, access to, and quality of healthcare services, and risk of maternal and infant mortality and morbidity may mean that women in low- and middle-income countries assign emphasis on domains of pregnancy-related anxiety that differ from women in high-income countries [83]. Moreover, the importance attached to these domains may change over the course of pregnancy. Future studies are required to determine if the association between concerns about the health of the fetus and preterm birth holds true with different samples in the same country and different countries. More research is required to establish the extent to which interventions provided to mothers who develop concerns about fetal health, could contribute to reducing preterm birth risk. Further research is also required to understand pregnancy-related anxiety more comprehensively since it concerns women's physical and emotional experiences of pregnancy, making it a unique and context-specific anxiety [84–86].

Another significant challenge is the lack of cross-culturally valid screening and diagnostic instruments for perinatal depressive symptoms particularly for use during the antepartum period [12]. The pregnancy-related anxiety scale we used did not include domains related to gender inequity, role of a pregnant woman in the household, socio-culturally specific experiences of pregnancy-related anxiety, or other contextual factors that may impact the experience of anxiety during pregnancy. Consequently, there is a need to develop pregnancy-related anxiety tools that accurately capture the complex nature of pregnancy-related anxiety for women living in various low- and middle-income countries [87, 88]. There is a paucity of literature about the realities and perspectives of women in low-income and middle-income countries experiencing pregnancy-related anxiety [87].

Finally, our finding of changes in psychosocial measures (rather than one-time assessments) suggests that women experiencing psychosocial distress should be screened more than once during pregnancy. Furthermore, it is also important to explore both the co-morbidities and the cumulative effects of various types of psychosocial distress to understand the true nature of the interactions between perceived stress, antenatal depressive symptoms, pregnancy-related anxiety, and their impacts on preterm birth. Lastly, research is also required to determine the time points during pregnancy, where screening for antenatal depressive symptoms and perceived stress, best determine change scores that may be predictive of preterm birth. Screening for various forms of psychosocial distress more than once during pregnancy has significant public health implications in terms of improving pregnancy experiences as well as improving pregnancy outcomes for both mothers and infants, populations that have generally been rendered vulnerable by societal structures in various resource-poor settings [45, 89, 90].

## Limitations

One limitation to this study is the relatively small sample size due to the pilot nature of our study. As such, the results of this study should be interpreted with caution as these results cannot be generalized. Secondly, since last menstrual period was self-reported, our data may be subject to recall bias. Third, there were several psychosocial factors that impact stress that were not measured including sleep, neighborhood disadvantage, lack of partner support, and life-

course variations, including precarious housing [19, 53]. Therefore, there could have been the limitation of residual confounding due to several unmeasured factors. Among the 24 women in our sample with preterm birth, five delivered before the third trimester. Additionally, another five women were lost to follow-up and not included in our sample due to giving birth prior to the third trimester. Therefore, the rate of preterm birth in our sample may be an underestimation. The second limitation of our study is that once our participants were screened, we did not complete clinical interviews to formally diagnose our participants with either depression or anxiety.

## Conclusion

The results of our study indicate that there may be a relationship between perceived stress and antenatal depressive symptoms and preterm birth. Women's anxiety about fetal heath was a marginally significant predictor of preterm birth. Our results must be interpreted with caution due to the small number of women in our stud that gave birth preterm. Further studies are required to determine if health care providers need to reorganize their care practices to address maternal concerns about fetal health early in pregnancy and monitor changes in antenatal depressive symptoms during pregnancy to identify women at risk of preterm birth in low- and middle-income settings.

## Acknowledgments

Dr. Shahirose Sadrudin Premji (premjis@yorku.ca) is the lead contact for the Maternal-infant Global Health Team.

Maternal Infant Global Health Team (MiGHT) Collaborators in Research members (alphabetical): Naureen Akber Ali[1], Neelofur Babar[7], Christine Dunkel Schetter[8], Fazila Faisal[1], Farooq Ghani[9], Nasreen Ishtiaq[1], Nigar Jabeen[10], Arshia Javed[11], Imtiaz Jehan[12], Fouzia Karim[1], Rabia Khoja[1], Nicole Letourneau[13,14,15,16], Mohamoud Merali[17], Qamarunissa Muhabat[10], Joseph Wangira Musana[18], Sidrah Nausheen[19], Christine Okoko[20], Almina Pardhan[21], Erum Saleem[11], Pauline Samia[22], and Salima Sulaiman[23]. We wish to express our sincere appreciation to the entire team!

[7]Department of Obstetrics & Gynaecology, The Aga Khan Hospital for Women, Karimabad, Karachi, Pakistan

[8]Department of Psychology, University of California Los Angeles, Los Angeles, California[9]Department of Pathology and Laboratory Medicine, Aga Khan University Hospital, Karachi, Pakistan

[10]Department of Obstetrics and Gynaecology, Aga Khan University Hospital, Hyderabad, Pakistan

[11]Deaapartment of Obstetrics and Gynaecology, Aga Khan University Hospital, Garden, Karachi, Pakistan

[12]Department of Community Health Sciences, Aga Khan University, Karachi, Pakistan

[13]Department of Pediatrics, University of Calgary, Calgary, Alberta, Canada

[14]Owerka Centre, Alberta Children's Hospital Research Institute, University of Calgary, Calgary, Alberta, Canada

[15]Faculty of Nursing, University of Calgary, Calgary, Alberta, Canada

[16]Department of Psychiatry, University of Calgary, Calgary, Alberta, Canada

[17]Department of Counselling and Clinical Psychology, Aga Khan University Hospital, Nairobi, Kenya

[18]Department of Gynecology, Aga Khan University Hospital, Nairobi, Kenya

[19]Department of Obstetrics and Gynaecology, Aga Khan University Hospital, Kharadar, Karachi, Pakistan

[20]School of Nursing and Midwifery, Aga Khan University, Nairobi, Kenya

[21]Institute for Educational Development, Aga Khan University, Karachi, Pakistan

[22]Department of Pediatrics and Child Health, Aga Khan University Hospital, Nairobi, Kenya

[23]Faculty of Nursing, University of Toronto, Toronto, Canada

## Author Contributions

**Conceptualization:** Aliyah Dosani, Shahirose Sadrudin Premji, Kiran Shaikh.

**Data curation:** Sharifa Lalani, Shahirose Sadrudin Premji, Kiran Shaikh.

**Formal analysis:** Ntonghanwah Forcheh, Shahirose Sadrudin Premji, Ilona S. Yim.

**Funding acquisition:** Shahirose Sadrudin Premji, Kiran Shaikh.

**Investigation:** Ntonghanwah Forcheh, Shahirose Sadrudin Premji.

**Methodology:** Aliyah Dosani, Ntonghanwah Forcheh, Shahirose Sadrudin Premji, Kiran Shaikh, Ilona S. Yim.

**Project administration:** Sharifa Lalani, Shahirose Sadrudin Premji.

**Resources:** Shahirose Sadrudin Premji, Kiran Shaikh.

**Supervision:** Shahirose Sadrudin Premji, Ilona S. Yim.

**Validation:** Aliyah Dosani, Ntonghanwah Forcheh, Shahirose Sadrudin Premji, Sana Siddiqui, Kiran Shaikh, Ayesha Mian, Ilona S. Yim.

**Visualization:** Shahirose Sadrudin Premji.

**Writing – original draft:** Sharifa Lalani, Aliyah Dosani, Ntonghanwah Forcheh, Shahirose Sadrudin Premji, Sana Siddiqui, Kiran Shaikh, Ayesha Mian, Ilona S. Yim.

**Writing – review & editing:** Sharifa Lalani, Aliyah Dosani, Ntonghanwah Forcheh, Shahirose Sadrudin Premji, Sana Siddiqui, Kiran Shaikh, Ayesha Mian, Ilona S. Yim.

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
