## [Decision Letter · Decision Letter 0]

7 Jan 2021

PONE-D-20-33144

Perceived stress may mediate the relationship between antenatal depressive symptoms and preterm birth: a pilot observational cohort study

PLOS ONE

Dear Dr. Dosani,

Thank you for submitting your manuscript to PLOS ONE. After careful consideration, we feel that it has merit but does not fully meet PLOS ONE’s publication criteria as it currently stands. Therefore, we invite you to submit a revised version of the manuscript that addresses the points raised during the review process.

We look forward to receiving your revised manuscript.

Kind regards,

Kelli K Ryckman

Academic Editor

PLOS ONE

Journal Requirements:

2.We note that you have indicated that data from this study are available upon request. PLOS only allows data to be available upon request if there are legal or ethical restrictions on sharing data publicly. For information on unacceptable data access restrictions, please see http://journals.plos.org/plosone/s/data-availability#loc-unacceptable-data-access-restrictions.

3. One of the noted authors is a group or consortium [Maternal-infant Global Health

Team (MiGHT) Collaborators in Research]. In addition to naming the author group, please list the individual authors and affiliations within this group in the acknowledgments section of your manuscript. Please also indicate clearly a lead author for this group along with a contact email address.

Reviewers' comments:

Reviewer's Responses to Questions

**Comments to the Author**

1. Is the manuscript technically sound, and do the data support the conclusions?

Reviewer #1: Partly

2. Has the statistical analysis been performed appropriately and rigorously? 

Reviewer #1: No

3. Have the authors made all data underlying the findings in their manuscript fully available?

Reviewer #1: Yes

4. Is the manuscript presented in an intelligible fashion and written in standard English?

Reviewer #1: No

5. Review Comments to the Author

Reviewer #1: Reviewer comments for authors

Thank you for the opportunity to review the manuscript titled “Perceived stress may mediate the relationship between antenatal depressive symptoms and preterm birth: a pilot observational cohort study.” This study expands the understanding on the interactions between maternal depression, stress and preterm birth. Despite this contribution, I believe there are opportunities to further strengthen this manuscript. I will share these in the order that they appear.

1. Abstract:

• Line # 31: Please consider making low risk hyphenated.

• Line # 37: Please consider removing “as” after insignificant.

• Line # 38: Please consider removing “the effects of”

• Line # 44: T of “there needs to be capitalized.

• Please add one sentence about the significance of the study in the abstract.

2. Introduction:

• Line # 54: Please consider re-wording this sentence. The way it starts with “at 18.9%” is not appropriate.

• Introduction was not written in the descriptive manner and was very vague. I think authors of this paper should conduct a detailed literature review to strengthen the introduction. They need to build up the case on why this study was needed. There were several sentences in which evidence was not clearly stated. Please consider provide more details about consistent and inconsistent findings of the previous literature.

• Line # 87: Interpretation of odds needs to be revised. Odds needs to be represented in the change in percentage. For instance, 170% higher odds in this case or 2.70 times more likely.

• Line # 88-89: The sentence “In addition, high .depression scores.” It seems very confusing.

• I suggest adding a few compelling statistics related to psychosocial risk factors of preterm birth. Please look at this article (available at https://www.sciencedirect.com/science/article/abs/pii/S1876201820305499) to strengthen the background of your study.

3. Methods:

• Did authors of this study attempted to calculate the minimum sample size and priori power analysis?

• Please consider adding more details in the survey instruments. For example, PSS scores are obtained by reversing responses (e.g., 0 = 4, 1 = 3, 2 = 2, 3 = 1 & 4 = 0) to the four positively stated items (items 4, 5, 7, & 8) and then summing across all scale items.

• How many items were there in the demographic block?

• Please consider calculating the response rate (249/300 = 83%) somewhere in the text.

• Methodology was not clearly described. For example, how continuous and categorical variables will be reported. For example, continuous variables will be reported as means and standard deviation; categorical variables as frequency and percentages. How the comparison across categorical variables of such paired sample was conducted? In addition, Hierarchal regression modelling (HRM) building process needs more explanation. What variables were entered across different levels of model?

• Was the assumption of normality investigated? Were the assumptions of HRM investigated?

4. Results:

• Results are not clearly stated. Which test they used to calculate percentage change in depression across time 1 and 2? Data provided in lines # 223-226, which statistical test was used? Please also report the p values.

• I suggest moving the research questions right at the end of introduction, where aim of the study was stated.

• Please consider adding one more table, which will be a descriptive table outlining the sample characteristics. This table will be the table 1 of your study. This is the standard practice. While reporting your sample characteristics, please report the measures of dispersion with the means.

• In the HRM, please report the delta R2 so readers would know how prediction of the dependent or outcome variable changed after adding independent variables in each block of HRM. This is the main purpose of HRM, which was not fully investigated.

5. Discussion

• What is the possible explanation of women with extended family and high educational status having lower risk of preterm deliveries?

• There are still some unmeasured factors which were not explored in this study. I suggest reading this recently published article (https://www.sciencedirect.com/science/article/abs/pii/S1876201820305499) and acknowledge the limitation of residual confounding due to several unmeasured factors.

• Please state the limitation of lack of generalizability of the results explicitly.

• Another limitation can be recall or self-reporting bias. Since the LMP was self-reported.

• I see authors discuss the consistent findings. I am sure there are several pieces of evidence which yield contrasting findings. Please discuss those too.

• With a lag of 3 years, there are latest research out there, which need to be discussed. Discussion section needs a significant strengthening with more recent data.

• Please discuss the public health implications too.

Overall feedback:

• This manuscript needs significant improvements in all sections of the manuscript. There are several gaps which need to be fixed to aid in understanding.

6. PLOS authors have the option to publish the peer review history of their article (what does this mean?). If published, this will include your full peer review and any attached files.

Reviewer #1: No

---

## [Author Response · Author response to Decision Letter 0]

20 Feb 2021

Editor Feedback Responses

We thank the editor for this comment. We have ensured that our manuscript meets PLOS ONE’s style requirements including those for file naming. The templates shared with us were very helpful. Thank you!

2.We note that you have indicated that data from this study are available upon request. PLOS only allows data to be available upon request if there are legal or ethical restrictions on sharing data publicly. For information on unacceptable data access restrictions, please see http://journals.plos.org/plosone/s/data-availability#loc-unacceptable-data-access-restrictions.

We will update your Data Availability statement on your behalf to reflect the information you provide. We thank the editor for this comment. Unfortunately, we do not have ethical clearance to share the data within a repository. Due to the confidential nature and personal data collected and the relatively small sample size sharing even de-identified data poses a risk that individual participants could be identified. It is of utmost importance that we protect the vulnerable population that agreed to participate in our study, especially in a social context like Pakistan, where mental health issues contribute to stigmatization within families and communities and which could cause considerable harm to our study participants. Please contact Saima Ejaz (saima.ejaz@aku.edu), a member of the Aga Khan University Institutional Review Board for further information. 

3. One of the noted authors is a group or consortium [Maternal-infant Global Health

Team (MiGHT) Collaborators in Research]. In addition to naming the author group, please list the individual authors and affiliations within this group in the acknowledgments section of your manuscript. Please also indicate clearly a lead author for this group along with a contact email address. We thank the review for this suggestion. We have clearly indicated a lead author for MiGHT along with a contact email address on the title page. We have also listed the institutional affiliations for the individuals listed in the acknowledgements section.

Reviewer Feedback 

1. Abstract:

• Line # 31: Please consider making low risk hyphenated.

• Line # 37: Please consider removing “as” after insignificant.

• Line # 38: Please consider removing “the effects of”

• Line # 44: T of “there needs to be capitalized.

• Please add one sentence about the significance of the study in the abstract. We thank the reviewer for these suggestions for improvement to our abstract.

We have made the term low-risk hyphenated

We have removed the word “as” after insignificant

We respectfully disagree with this suggestion as it will change the meaning of the sentence as intended

We have capitalized the T in there.

We have added the following sentence to the abstract regarding the significance of this: “This is the first study of its kind to be conducted in Pakistan.”

2. Introduction:

• Line # 54: Please consider re-wording this sentence. The way it starts with “at 18.9%” is not appropriate.

• Introduction was not written in the descriptive manner and was very vague. I think authors of this paper should conduct a detailed literature review to strengthen the introduction. They need to build up the case on why this study was needed. There were several sentences in which evidence was not clearly stated. Please consider provide more details about consistent and inconsistent findings of the previous literature.

• Line # 87: Interpretation of odds needs to be revised. Odds needs to be represented in the change in percentage. For instance, 170% higher odds in this case or 2.70 times more likely.

• Line # 88-89: The sentence “In addition, high .depression scores.” It seems very confusing.

• I suggest adding a few compelling statistics related to psychosocial risk factors of preterm birth. Please look at this article (available at https://www.sciencedirect.com/science/article/abs/pii/S1876201820305499) to strengthen the background of your study. We thank the reviewer for the comments provided to strengthen the introduction.

Line #54” - we have reworded this sentence.

We have completed a thorough review of the literature and re-written the introduction to build the case on why this study was needed. We have added 2 paragraphs providing more details about the findings of the previous literature.

We have represented the odds as the change in percentage as requested.

We have changed the wording of this sentence to increase clarity. It now reads “In addition, women who experienced consistently low or high depression scores did not have a higher odds of preterm birth as compared to women who experienced a decrease in depression scores.”

We have added a few compelling statistics related to psychosocial risk factors and preterm birth.

3. Methods:

• Did authors of this study attempted to calculate the minimum sample size and priori power analysis?

• Please consider adding more details in the survey instruments. For example, PSS scores are obtained by reversing responses (e.g., 0 = 4, 1 = 3, 2 = 2, 3 = 1 & 4 = 0) to the four positively stated items (items 4, 5, 7, & 8) and then summing across all scale items.

• How many items were there in the demographic block?

• Please consider calculating the response rate (249/300 = 83%) somewhere in the text.

• Methodology was not clearly described. For example, how continuous and categorical variables will be reported. For example, continuous variables will be reported as means and standard deviation; categorical variables as frequency and percentages. How the comparison across categorical variables of such paired sample was conducted? In addition, Hierarchal regression modelling (HRM) building process needs more explanation. What variables were entered across different levels of model?

• Was the assumption of normality investigated? Were the assumptions of HRM investigated? We thank the reviewer for these comments.

We did not calculate the sample size or power analyses. Please note that this is a secondary analyses of data. We have added a statement to this effect in the study procedures and participants section.

We have added in more details about the survey instruments used. The following sentence has been added in the description of the pregnancy-related anxiety scale: “The first five items are scored on a scale of 1 to 4 while items 6-10 are rated as never, sometimes, most of the time, or all of the time. While there is no established cut-off for the scale, three or more positive responses in column four has been used to suggest the presence of anxiety.”

The following sentence has been added to the description of the Edinburgh Postnatal Depression Scale: “Items 1 to 4 are scored from 0-3, while items 5 – 10 are scored from 3-0. The numbers are then tallied.”

The following sentence has been added to the description of the perceived stress scale: “Perceived stress scores are obtained by reversing responses (e.g., 0 = 4, 1 = 3, 2 = 2, 3 = 1 & 4 = 0) to the four positively stated items (items 4, 5, 7, & 8) and then summing across all scale items.”

We have added the following sentence to the description of demographic information obtained: “A total of 36 items were explored in the demographic survey.”

The response rate is clearly explained in lines 208.

We have added in the following sentence in the data analysis and statistical methods section: “Descriptive data was explored with continuous variables reported as means and standard deviation and categorical variables as frequency and percentages.” In lines 240 – 242 we indicate how the comparison across categorical variables of the paired sample was conducted in terms of t-tests and Pearson Correlations. In lines 242 – 253 we clearly describe the process used in our hierarchical regression modeling. We respectfully offer that adding in more details would lengthen this section unnecessarily. 

Yes, the assumptions of normality were investigated and yes, the assumptions of HRM were investigated. We have added in statements reflecting this in the data analysis and statistical methods section.

4. Results:

• Results are not clearly stated. Which test they used to calculate percentage change in depression across time 1 and 2? Data provided in lines # 223-226, which statistical test was used? Please also report the p values.

• I suggest moving the research questions right at the end of introduction, where aim of the study was stated.

• Please consider adding one more table, which will be a descriptive table outlining the sample characteristics. This table will be the table 1 of your study. This is the standard practice. While reporting your sample characteristics, please report the measures of dispersion with the means.

• In the HRM, please report the delta R2 so readers would know how prediction of the dependent or outcome variable changed after adding independent variables in each block of HRM. This is the main purpose of HRM, which was not fully investigated. We thank the reviewer for these comments.

We have added in the requested details in the descriptive section of the results: “The mean baseline score, change in means (D), standard error (SE) and two-sided paired t-tests, and p-value (p) for pregnancy-related anxiety, antenatal depressive symptoms, and perceived stress were 11.8 (D=0.418, SE=0.328, p=0.204), 5.9 (D= -0.932, SE 0.324, p=0.004), and 15.1 (D=-0.386, SE=0.433, p=0.375).”

We thank the reviewer for this suggestion. While the aims of our study which are presented at the end of the introduction clearly align with our research questions, we have deliberately broken down our results section by research question to demonstrate how we have aligned the results with each question. We have done this to help organize our results as we have many research questions that were answered in this work.

We have added in one more table outlining the descriptive statistics. This is now Table 1 as suggested.

We have included the following sentences just before Table 2 and 3 as requested: “The adjusted R-squared for the final model was 11.8% indicating a low predictive power.” And “The adjusted R-squared for the final model was 12.4% indicating a low predictive power.”

5. Discussion

• What is the possible explanation of women with extended family and high educational status having lower risk of preterm deliveries?

• There are still some unmeasured factors which were not explored in this study. I suggest reading this recently published article (https://www.sciencedirect.com/science/article/abs/pii/S1876201820305499) and acknowledge the limitation of residual confounding due to several unmeasured factors.

• Please state the limitation of lack of generalizability of the results explicitly.

• Another limitation can be recall or self-reporting bias. Since the LMP was self-reported.

• I see authors discuss the consistent findings. I am sure there are several pieces of evidence which yield contrasting findings. Please discuss those too.

• With a lag of 3 years, there are latest research out there, which need to be discussed. Discussion section needs a significant strengthening with more recent data.

• Please discuss the public health implications too.

Overall feedback:

• This manuscript needs significant improvements in all sections of the manuscript. There are several gaps which need to be fixed to aid in understanding. We thank the reviewer for these comments.

We have provided possible explanations of why women with extended families and higher educational status have lower risk of preterm birth in the first paragraph of the discussion.

We have included this paragraph in our limitations: “Third, there were several psychosocial factors that impact stress that were not measured including neighborhood disadvantage, lack of partner support, and life-course variations, including precarious housing (Batra et al., 2020). Therefore, there could have been the limitation of residual confounding due to several unmeasured factors.”

We have explicitly stated the lack of generalizability of our results.

We have indicated recall bias as a limitation in our study.

We have added in the following sentence in our discussion that highlights the main inconsistent finding: “Our results differ from those of Glynn et al. (2008) who found that pregnant women who exhibit a decrease in perceptions of psychosocial distress and dampening of biological stress responses in the late second trimester were less likely to deliver preterm.” 

We have added information from these latest research articles to our discussion:

1. Batra K, Pharr J, Olawepo JO, Cruz P. Understanding the multidimensional trajectory of psychosocial maternal risk factors causing preterm birth: A systematic review. Asian journal of psychiatry. 2020 Oct 15:102436.

2. Hung HY, Su PF, Wu MH, Chang YJ. Status and related factors of depression, perceived stress, and distress of women at home rest with threatened preterm labor and women with healthy pregnancy in Taiwan. Journal of Affective Disorders. 2021 Feb 1;280:156-66.

3. Tomfohr-Madsen L, Cameron EE, Dunkel Schetter C, Campbell T, O'Beirne M, Letourneau N, Giesbrecht GF. Pregnancy anxiety and preterm birth: The moderating role of sleep. Health psychology. 2019 Nov;38(11):1025.

4. Weis KL, Walker KC, Chan W, Yuan TT, Lederman RP. Risk of preterm birth and newborn low birthweight in military women with increased pregnancy-specific anxiety. Military medicine. 2020 May;185(5-6):e678-85.

5. Dadi AF, Miller ER, Bisetegn TA, Mwanri L. Global burden of antenatal depression and its association with adverse birth outcomes: an umbrella review. BMC Public Health. 2020 Dec 1;20(1):173.

6. Dadi AF, Miller ER, Mwanri L. Antenatal depression and its association with adverse birth outcomes in low and middle-income countries: A systematic review and meta-analysis. PloS one. 2020 Jan 10;15(1):e0227323.

7. Ghimire U, Papabathini SS, Kawuki J, Obore N, Musa TH. Depression during pregnancy and the risk of low birth weight, preterm birth and intrauterine growth restriction-an updated meta-analysis. Early Human Development. 2021 Jan 1;152:105243.

We have added the following sentence regarding public health implications in the last paragraph of the discussion: “Screening for various forms of psychosocial distress more than once during pregnancy has significant public health implications in terms of improving pregnancy experiences as well as improving pregnancy outcomes for both mothers and infants, populations that have generally been rendered vulnerable by societal structures in various resource-poor settings.

We hope the changes that we have incorporated have significantly improved our manuscript. We have addressed the gaps to aid in understanding as indicated by the reviewer.

---

## [Decision Letter · Decision Letter 1]

29 Mar 2021

PONE-D-20-33144R1

Perceived stress may mediate the relationship between antenatal depressive symptoms and preterm birth: a pilot observational cohort study

PLOS ONE

Dear Dr. Dosani,

Thank you for submitting your manuscript to PLOS ONE. After careful consideration, we feel that it has merit but does not fully meet PLOS ONE’s publication criteria as it currently stands. Therefore, we invite you to submit a revised version of the manuscript that addresses the points raised during the review process.

Please address the reviewers minor comments in the attached document.

We look forward to receiving your revised manuscript.

Kind regards,

Kelli K Ryckman

Academic Editor

PLOS ONE

Journal Requirements:

Reviewers' comments:

Reviewer's Responses to Questions

**Comments to the Author**

1. If the authors have adequately addressed your comments raised in a previous round of review and you feel that this manuscript is now acceptable for publication, you may indicate that here to bypass the “Comments to the Author” section, enter your conflict of interest statement in the “Confidential to Editor” section, and submit your "Accept" recommendation.

Reviewer #1: All comments have been addressed

2. Is the manuscript technically sound, and do the data support the conclusions?

Reviewer #1: Yes

3. Has the statistical analysis been performed appropriately and rigorously? 

Reviewer #1: Yes

4. Have the authors made all data underlying the findings in their manuscript fully available?

Reviewer #1: Yes

5. Is the manuscript presented in an intelligible fashion and written in standard English?

Reviewer #1: Yes

6. Review Comments to the Author

Reviewer #1: (No Response)

7. PLOS authors have the option to publish the peer review history of their article (what does this mean?). If published, this will include your full peer review and any attached files.

Reviewer #1: No

---

## [Author Response · Author response to Decision Letter 1]

2 Apr 2021

In response to the sample size justification, authors indicated that the study involved secondary analysis. However, there is no mention about it in the manuscript. In-fact, authors have described sample recruitment and prospective design. I am not sure how this study is a secondary analysis. More information will be appreciated. 

We thank the reviewer for this comment. We have added in the following sentence in the study design section: “This is a secondary analysis of data that were collected between October 2015 and July 2016.”

I am still not understanding the actual meaning of this sentence: “In addition, women who experienced consistently low or high depression scores did not have a higher odd of preterm birth as compared to women who experienced a decrease in depression scores.” I am not sure what authors are trying to convey?

We thank the reviewer for this comment. We have deleted the sentence in question to improve clarity and flow of the manuscript.

Tables can be improved per journal standard.

We thank the reviewer for this comment. We have taken the information from the journal’s author guidelines and have reformatted our tables accordingly.

https://journals.plos.org/plosone/s/submission-guidelines

https://journals.plos.org/plosone/s/tables

---

## [Editor Report · Decision Letter 2]

19 Apr 2021

Perceived stress may mediate the relationship between antenatal depressive symptoms and preterm birth: a pilot observational cohort study

PONE-D-20-33144R2

Dear Dr. Dosani,

We’re pleased to inform you that your manuscript has been judged scientifically suitable for publication and will be formally accepted for publication once it meets all outstanding technical requirements.

Kind regards,

Kelli K Ryckman

Academic Editor

PLOS ONE
---

## [Editor Report · Acceptance letter]

21 Apr 2021

PONE-D-20-33144R2 

Perceived stress may mediate the relationship between antenatal depressive symptoms and preterm birth: a pilot observational cohort study  

Dear Dr. Dosani:

I'm pleased to inform you that your manuscript has been deemed suitable for publication in PLOS ONE. Congratulations! Your manuscript is now with our production department. 

Kind regards, 

on behalf of

Dr. Kelli K Ryckman 

Academic Editor

PLOS ONE